# Low-Cost 3D Virtual and Dynamic Reconstruction Approach for Urban Forests: The Mesiano University Park

Chiara Chioni *, Anna Maragno, Angelica Pianegonda, Marco Ciolli, Sara Favargiotti and Giovanna A. Massari

Department of Civil, Environmental and Mechanical Engineering (DICAM), University of Trento, 38123 Trento, Italy; anna.maragno@unitn.it (A.M.); angelica.pianegonda@unitn.it (A.P.); marco.ciolli@unitn.it (M.C.); sara.favargiotti@unitn.it (S.F.); giovanna.massari@unitn.it (G.A.M.)
* Correspondence: chiara.chioni@unitn.it

**Abstract:** Urban forests, parks, and gardens are fundamental components of urban sustainability, resilience, and regenerative dynamics. Designers, architects, and landscape architects could smartly manage these dynamic ecosystems if efficiently provided with design-oriented digital tools, technologies, and techniques. However, practitioners lack knowledge and standardized procedures for their uses. The rise of low-cost sensors to generate 3D data (e.g., point clouds) in forestry can also effectively support monitoring, analysis, and visualization purposes for greenery in urban contexts. Adopting an interdisciplinary approach—involving the fields of forestry, geomatics, and computer science—this contribution addresses these issues and proposes a low-cost workflow for 3D virtual reconstructions of urban forests to support information management activities and thus landscape architecture applications. By connecting a wide range of methods (i.e., spherical photogrammetry, point cloud modeling), tools (i.e., 360° camera, tablet with lidar sensor), and software (i.e., Agisoft Metashape, CloudCompare, Autodesk AutoCAD), the proposed workflow is defined and tested in the development of dynamic virtual representations for a plot of the Mesiano University park in Trento (Italy). Finally, comparing acquisition, processing, and elaboration methodologies and their results, the possibility of developing digital twins of urban forests is envisioned.

**Keywords:** landscape representation; digital landscape; low-cost 3D; urban forest survey; spherical photogrammetry; point cloud modeling

## 1. Introduction

Green infrastructure—such as urban forests, parks, and gardens—are fundamental components of urban sustainability and resilience [1], offering to cities several ecosystem services (namely provisioning, regulation and maintenance, and cultural services) [2] with a regenerative perspective [3]. Therefore, attention is growing toward the smart management, planning, and design of these dynamic ecosystems, requiring a multidisciplinary approach [4]. To support forestry applications in urban contexts, using new digital technologies, the prominent field could be landscape architecture [5] in dialogue with geomatics and computer science. Indeed, designers, architects, and landscape architects can potentially tackle the main steps of information management activities—data acquisition and processing, data elaboration, and results' visualization—about urban trees and green spaces to support their planning and design, but nowadays, they are experiencing a lack of knowledge about methodologies and tools, from one side, and of standardized procedures, from the other side, to address the related issues.

Currently, the procedures for acquiring information on urban greenery are mostly carried out following forest inventories' practices. Such inventories are crucial for various purposes: (i) forest management, by assessing forests' composition, structure, and the overall ecological health of the ecosystem [6,7]; (ii) resources' assessment [7–9], by estimating

timber volume, biomass, and carbon stocks; (iii) biodiversity conservation, by identifying and monitoring plant and animal species, and critical habitats [7]; (iv) environmental monitoring, by tracking changes in the forest capacity to improve air, water, and soil quality [7]; (v) decision making [10], by informing the development of green infrastructure policies, land-use planning, conservation strategies, and tree inventories; and (vi) community engagement [11], by fostering a sense of ownership, and raising awareness about ecological, social, cultural, and recreational services. Moreover, forest inventories are also helpful from a research and education perspective [12], by providing baseline data for ecological studies, climate change research, wildlife management, and educational programs. Traditional data acquisition systems are based on two separate procedures, inventory and mapping, but this division is a limitation [6]; for example, it does not allow changes in time and space to be observed. Digital procedures for data acquisition could overcome the limitations of traditional systems [13,14].

Given this framework, the purpose of this contribution is to identify a low-cost approach, complementary to traditional methods, for 3D virtual reconstructions of urban forests in order to eventually support information management activities and thus landscape architecture applications. Specifically, low cost refers to the following adopted in the proposed workflow: the relatively economic sensors, the mainly open-source/free license software and plugins (also generating open file formats), the simplicity and repeatability of the process that does not require highly specialized skills, and the time-saving procedures during the acquisition phase and in terms of computational power.

The structure of this paper is as follows: Section 2 presents an overview of the traditional survey framework and briefly introduces the current issue of 3D data in forestry. Section 3, after stating the aim of this paper and presenting the case study, describes the methods, tools, and software used in the data acquisition, processing, and elaboration steps of the proposed workflow. The results are summarized in Section 4 and then discussed in Section 5, with conclusions and the future outlook.

## 2. Background

This section first presents the conventional framework for forest surveys (Section 2.1) and then introduces the growing topic of digital 3D data in forestry (Section 2.2).

### 2.1. Conventional Framework for Forest Survey

A traditional, terrestrial urban forest survey can range from small groups of trees to large plots and can be conducted through systematic, stratified, or random sampling, with methods and tools that may vary depending on specific objectives, local conditions, and available resources [12,15]. Generally, the process involves defining objectives, selecting the study area, planning the survey, gathering the necessary equipment, conducting vegetation assessments, estimating timber volume [16], and assessing biodiversity [17]; it is essential to preliminarily consult the relevant literature, forestry experts, and/or local forestry agencies for specific guidelines and best practices in the studied region. Data and observation recording in a notebook and/or data sheets is crucial during fieldwork, as data analysis is conducted after completing the survey; finally, a comprehensive report or presentation is prepared to communicate the relevant findings to the stakeholders.

The main parameters that can be acquired through traditional methods are forests' composition, structure, and health; plant and animal biodiversity, with particular attention to rare, endangered, or invasive species (that can be indicators for forest health and habitat quality); and topography-related information [16]. Specifically, it is essential to account for each species' composition (i.e., grass, brush, and trees) by considering their presence in the study area; this parameter is helpful in forecasting future regenerations and in understanding if past (natural or artificial) regenerations have occurred. In the case of trees, it is important to measure—with a measuring tape or a caliper—the Diameter at Breast Height (DBH), namely at 1.3 m above the ground, and—with an hypsometer—the height to obtain information about their stand structure, potential growth, volume, and

biomass [16,18]; moreover, information about the crown (i.e., diameter and shape), that can be derived from the tree competition and the overall light availability, is useful to estimate the canopy cover [16,18].

The main limitations of traditional methods are in assessing dynamic processes and representing non-tree components; in fact, they are often better suited for static attributes (as tree dimensions and species composition), rather than capturing dynamic processes, mainly because they lack in defining precise spatial localizations and thus in identifying changes over time of both tree and non-tree components.

The approach proposed in Section 3, not intending to replace conventional techniques and the data resulting from their use in their entirety, specifically supports the acquisition of digital 3D data in forestry to partially overcome these issues and subsequently derive parameters that depend on geometric features.

### 2.2. The Advent of Digital 3D Data in Forestry

In the traditional field of forestry, the use of digital 3D data—together with the concept of 'virtual forest'—seems to be particularly relevant to investigate the inherently dynamic nature of forests (as well as to support spatial decision making and promote stakeholder engagement and public empowerment) and is indeed on the rise, especially for research, training, gaming, and forest planning applications [19].

In the framework of reality-based methods to develop digital forests, close-range remote sensing has changed in situ inventories [20,21], allowing for many previously impossible investigations, such as deriving information about stems (i.e., position, curvature, diameter), branches (i.e., angles, diameters, distances between branches), canopy (i.e., multispectral characteristics, size, length, area, volume distribution), and neighborhood (i.e., number of trees, size and relative position, dominance) [22]. In addition, various research studies have addressed the automation of segmentation processes and the derivation of relevant metrics for forest planning [23–27]; machine learning techniques have also been applied to help with this task [28].

Even if expensive and time-consuming terrestrial laser scanning still is the gold standard for ground-based 3D reconstructions in forestry, low-cost sensors have recently been raised as solid alternatives [29] using both the active sensing method, as Solid-State Lidar sensors in the latest Apple products [30,31], and passive sensing method, as 360° cameras for spherical photogrammetry [32]. Specifically, the cost of the mentioned instruments ranges from tens of thousands of euros for a Terrestrial Laser Scanner (TLS), just over a thousand euros for the latest iPhone or iPad, and from a few hundred up to a few thousand euros for a 360° camera. Overcoming the high acquisition costs, 3D point cloud models (i.e., the most direct measurement and a high-quality representation of real-world objects) of individual trees as well as forest ecosystems could be completely exploited—with their aesthetic, analytical, and generative potential [33]—through workflows in landscape and urban design also toward the development of 'forest digital twins' [34,35].

Indeed, the larger framework of this exploration is ongoing research investigating the development of a Territorial Digital Twin (TDT)—of which forest digital twins would be essential components—as a multiscale 3D digital copy of fragile landscapes to increase their resilience [36]. Specifically, even if the modelling of the natural environment has only recently started to be developed in urban digital twins (e.g., the GreenTwins project for Helsinki and Tallinn [37]), it seems crucial to address the rising importance of urban vegetation for the well-being of citizens.

### 3. Low-Cost Workflow for 3D Urban Forests: Tools and Methods

The methodological and operational proposal of this contribution investigates the integrated application of digital tools, technologies, and techniques in the landscape management, planning, and design of urban forests and parks to support information management activities [6]. In light of the studies already conducted on this issue [29,32,38], the research design aims specifically at developing a new low-cost workflow for 3D digital landscape

reconstructions through an interdisciplinary approach, integrating landscape planning and design with survey and digital modeling disciplines.

The workflow is 'low-cost' because, as mentioned, it uses low-cost sensors (in terms of the actual price of the tools compared to TLS) and it requires less time and resources during the data acquisition phase; in addition, it mostly requires software and plugins that are open-source or have free/educational licensing options, thus also generating open file formats.

Connecting a wide range of methods (i.e., spherical photogrammetry, point cloud modeling), tools (i.e., 360° camera, tablet with lidar sensor), and software (i.e., Agisoft Metashape version 1.7.0, CloudCompare version 2.12.4, Autodesk AutoCAD version 24.2.53.0), the proposed workflow is defined (Figure 1) and tested in a non-trivial, sufficiently self-contained, real-world applied research exploration: the development of dynamic virtual representations for a plot of the Mesiano University park in Trento (Italy). The result is a 3D digital model that would allow the acquisition of information at micro- (understory vegetation), meso- (single tree), and macro- (urban park or forest) spatial scales.

In order to maximize the information regarding the case study and to monitor forest parameters over time, the 3D digital survey was carried out in two phases [39]. The campaign was carried out on two different days: 21 February 2023, when the trees were bare and their wood framework was better detectable; and 22 June 2023, when the trees had foliage and the grass was vigorous and visible.

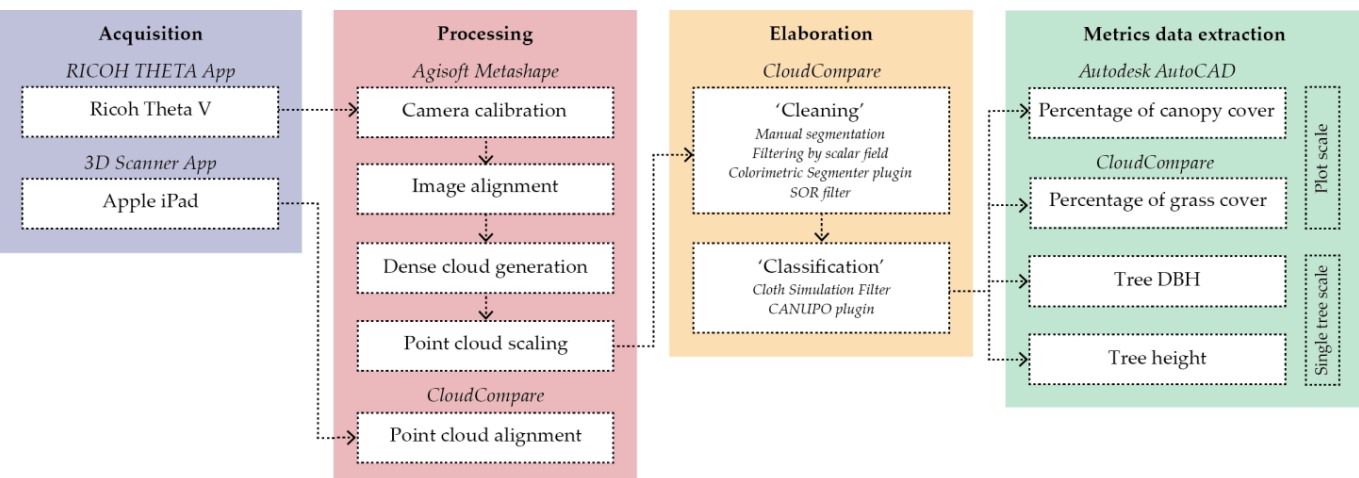

**Figure 1.** Workflow of the research exploration: the data collection with passive and active sensors; the data processing and elaboration with dedicated software to produce the 3D tree models; and finally, the extraction of the main forestry parameters.

### 3.1. Case Study: The Mesiano University Park in Trento (Italy)

The case study presented for our research exploration is located in the city of Trento (Italy), the capital of the same province, in the eastern Italian Alps. The city lies along the course of the Adige River and, being within a valley, is surrounded by two mountain ranges reaching up to 2000 m asl. Specifically, the study area is located within the park of the Department of Civil Environmental and Mechanical Engineering (DICAM) of the University of Trento, on the east hill of the valley (lat. 46.065231, long. 11.139920). The park covers an area of approximately 25,000 m² and mainly has conifers and deciduous trees.

Inside the test area (Figure 2), on the northeast side of the park, there are eleven trees, four *Platanus acerifolia* and seven *Pinus Sylvestris*, distributed in approximately 580 m² (about 18 × 32 m), consistent with the standard dimensions for forest surveys (i.e., 20 × 20 m). This urban green space has structurally different characteristics compared to 'natural' forest and thus has its own specific management, planning, and design questions. Indeed, in urban contexts, because of pruning and generally large distances between trees, the canopy and the branches can have a significant horizontal development, while in

'proper' forest, since the density of vegetation is higher and there are no pruning operations, trees mainly develop vertically, in order to reach the sunlight.

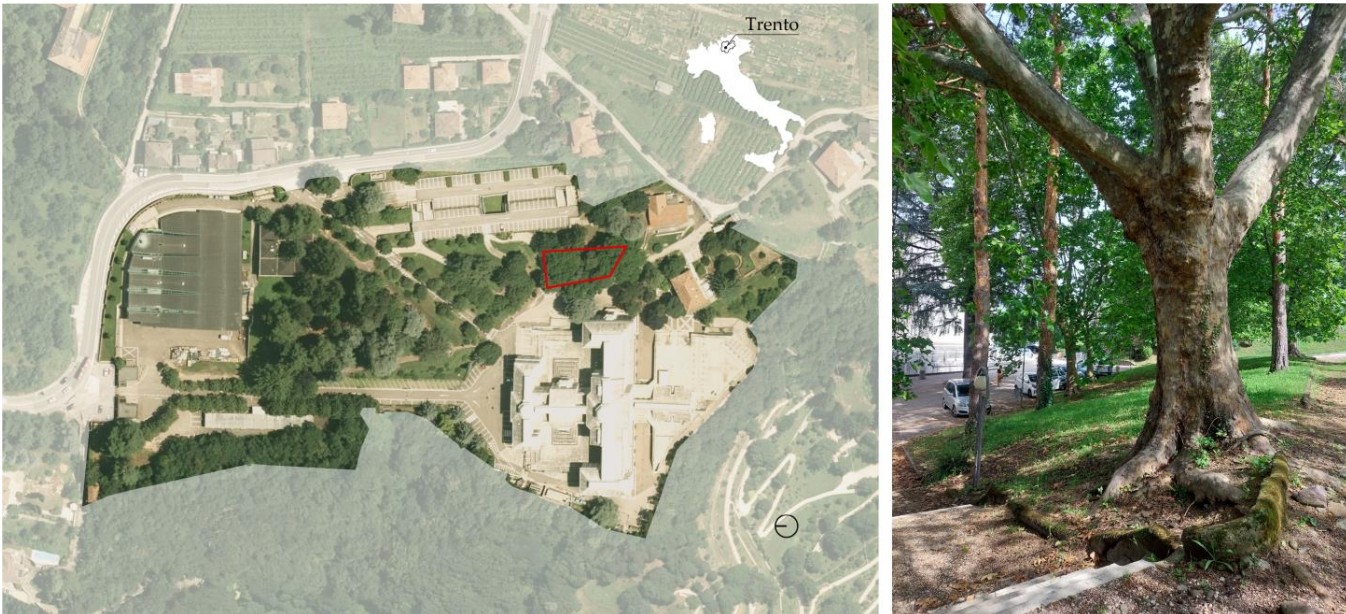

**Figure 2.** Case study location (red boundary) in the Mesiano University campus in Trento, Italy. Photo by the authors (C.C.).

A traditional survey was initially conducted to have a solid reference for the later comparison with the digital survey's methodology and output. Data about tree species, DBH, and height were recorded for each of the eleven trees (Table 1): a tree caliper and a digital hypsometer were used to acquire the relevant data, a field notebook and sheets were used to record observations, and chalk was used to mark trees and prevent double-counting or non-counting.

**Table 1.** Summary of the traditionally acquired data; DBH and height are average values from the measurements taken. For the nomenclature of the trees, see Figure 3.

| Tree | Species | DBH (m) | Height (m) |
|------|---------|---------|------------|
| A | *Platanus × acerifolia* | 0.77 | 21.7 |
| B | *Pinus sylvestris* | 0.63 | 23.8 |
| C | *Pinus sylvestris* | 0.46 | 23.8 |
| D | *Platanus × acerifolia* | 0.48 | 24.3 |
| E | *Pinus sylvestris* | 0.33 | 22.5 |
| F | *Pinus sylvestris* | 0.46 | 22.0 |
| G | *Pinus sylvestris* | 0.56 | 23.2 |
| H | *Platanus × acerifolia* | 0.51 | 27.2 |
| I | *Pinus sylvestris* | 0.58 | 26.8 |
| L | *Pinus sylvestris* | 0.53 | 23.3 |
| M | *Platanus × acerifolia* | 0.95 [1] | 25.8 |

[1] Since it was not possible to directly measure the diameter of the trunk with a caliper, the circumference (3 m) was measured with a tape and then the diameter calculated.

### 3.2. Data Acquisition

For data acquisition in the winter campaign, we conducted a terrestrial photogrammetric survey with the 360° camera Ricoh Theta V, combined with a lidar terrestrial survey using an Apple iPad. For the summer campaign, on the other hand, we only used the Ricoh Theta V, as the instrument's acquisition range of the iPad did not allow for the acquisition of tree crowns, thus not detecting significant differences.

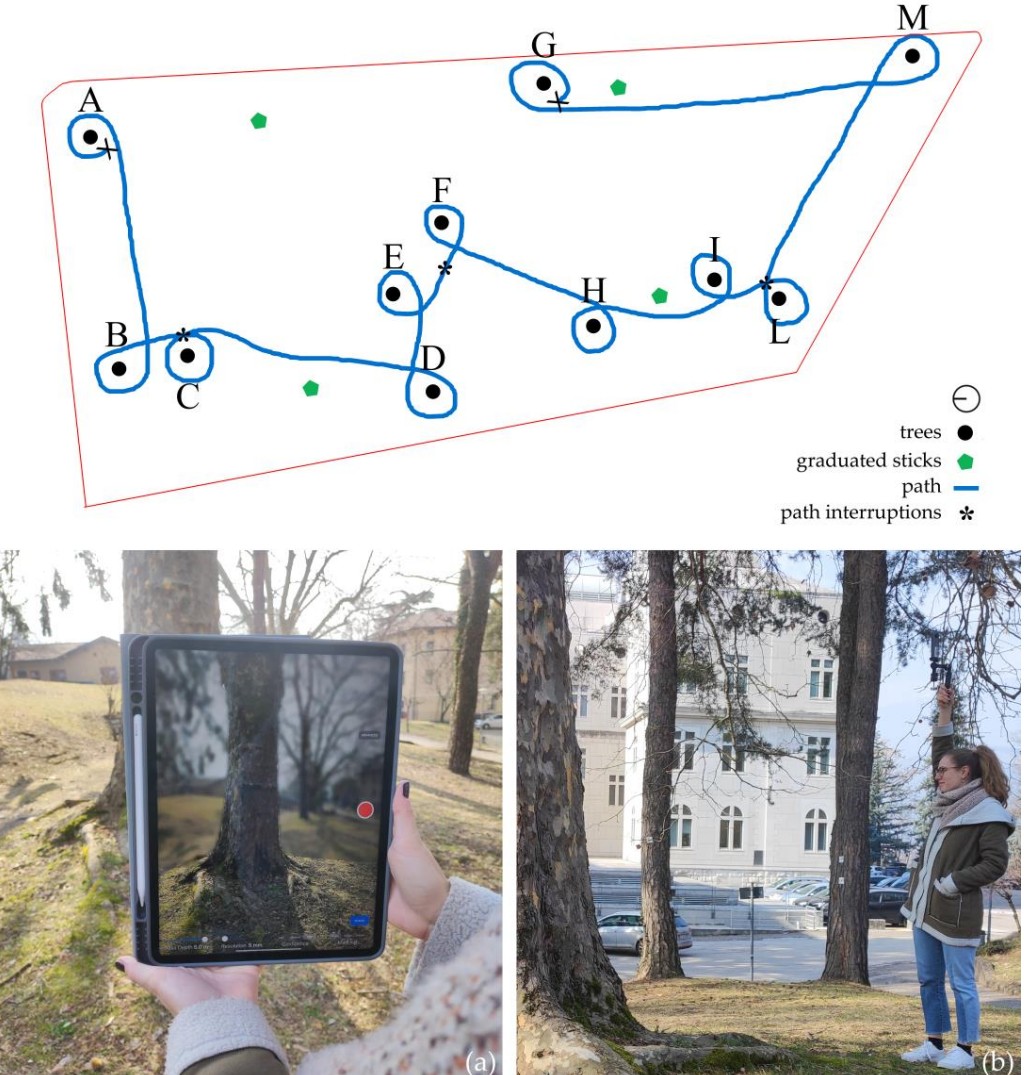

**Figure 3.** The data acquisition path (blue line) of both the used devices, Apple iPad (**a**) and Ricoh Theta V (**b**), on the study plot (red boundary); the black dots, named with capital letters, represent the tree positions; the asterisks (*) along the path indicate interruptions in iPad scans. Photos by the authors (A.M.).

In order to then be able to scale and align the 3D point clouds, prior to the acquisition of spherical video and scans, it was necessary to place several targets in the study area, in line with common 3D registration requirements [29]. During the winter and summer surveys, we used two different types of targets: in winter, we hung 25 coded-targets on tree trunks, while in summer, we employed four graduated sticks planted in the ground to form an approximately square area (about 15 × 15 m), sufficiently large in relation to the reference plot. In both cases, we measured the linear distances between the markers with a tape measure in order to later set up the scale of the 3D models. The choice to use different types and quantities of targets derives from the different surveying methodologies applied. With the Apple iPad, for reasons of device memory and acquisition range (approximately 5 m), it was necessary to divide the acquisition path into four segments, thus obtaining several point clouds that had to be aligned into one by means of coded-targets [29]; even if the coded-targets do not have georeferenced coordinates, the point cloud from the iPad was nevertheless roughly scaled and georeferenced later in the processing step (see Section 3.2), thanks to the embedded GNSS receiver of the device, having its theoretical precision on the order of meters. On the other hand, with the 360° camera, we captured the entire scene in

a single video, thus obtaining a single global point cloud that could be scaled and oriented with fewer markers, i.e., the graduated sticks.

The data acquisition with the Apple iPad (Figure 3a) was conducted using the 3D Scanner App (https://3dscannerapp.com/, accessed on 26 June 2023)—between the most suitable apps for this application [30]—with the 'LiDAR advanced' setting and the 'high resolution' mode (meaning max depth of 5 m, resolution of 5 mm, confidence set to high, masking set to off). Following the current best practices [40], the scans were acquired by slowly circling each tree in the plot, at a distance of 2/3 m from the trunk and 1.30/1.50 m above the ground, trying not to scan the same area twice. Since the acquisition was divided into four different scans, as mentioned, at the beginning of the new section, it was necessary to scan the last tree acquired in the previous scanning in order to allow the subsequent alignment.

With regard to the spherical video acquisitions with the Ricoh Theta V 360° camera (Figure 3b), they were acquired using the default settings (i.e., 4k, 29.97 fps) again by walking around the study plot, circling each tree slowly, at a distance of 2/3 m from the trunk and above the operator's head. The 'winter' video had 4:46 min of duration and the 'summer' video had 3:06 min of duration.

### 3.3. Data Processing

The data need to be processed, following a tailored procedure according to the acquisition mode, and converted into used formats for digital application (as 3D point clouds).

**Scan processing.** The 3D Scanner App used to scan also allows for directly and automatically processing the scans (in HD mode), meaning the addition of texture and detail. The generated textured 3D point clouds can then be exported in several formats; the 'XYZ color, space delimited (CloudCompare)' format was selected to optimize the compatibility with the open-source software CloudCompare (CC) (https://www.danielgm.net/cc/, accessed on 26 June 2023). Indeed, even if in principle the clouds neither need to be scaled nor georeferenced thanks to the embedded GNSS, the alignment of the different scans through the targets still needs to be performed in order to obtain a unique, single model (Figure 4a).

**Spherical photogrammetry.** Before proceeding with the processing, the spherical videos acquired with the 360° camera needed to be stitched with the RICOH THETA app (for desktop), version 3.19.4. The stitched videos can therefore be imported into the software Agisoft Metashape Professional (https://www.agisoft.com, accessed on 26 June 2023) that allows for splitting them into a number of frames per second; in our specific case, extracting 5 frames per second results in a total of 1720 frames from the winter video and 1119 frames from the summer video. Then, the spherical photogrammetry procedure followed the main steps and general recommendations for processing images from HDR panorama cameras: after setting the spherical camera model, first the photo alignment and then the dense point cloud building were performed, setting the 'high' accuracy. Finally, the 3D models were scaled by the bar scale method: using the known linear distances between the coded targets for the winter model, while using the linear distances between the graduated sticks for the summer model (Figure 4b).

After the processing, it appears evident that the iPad is severely limited by its range for this particular application; indeed, it cannot allow a 'complete' scan of the urban forest plot. For this reason, the following steps were performed only for the point clouds resulting from spherical photogrammetry, but in principle, they can be carried out also for the iPad's output.

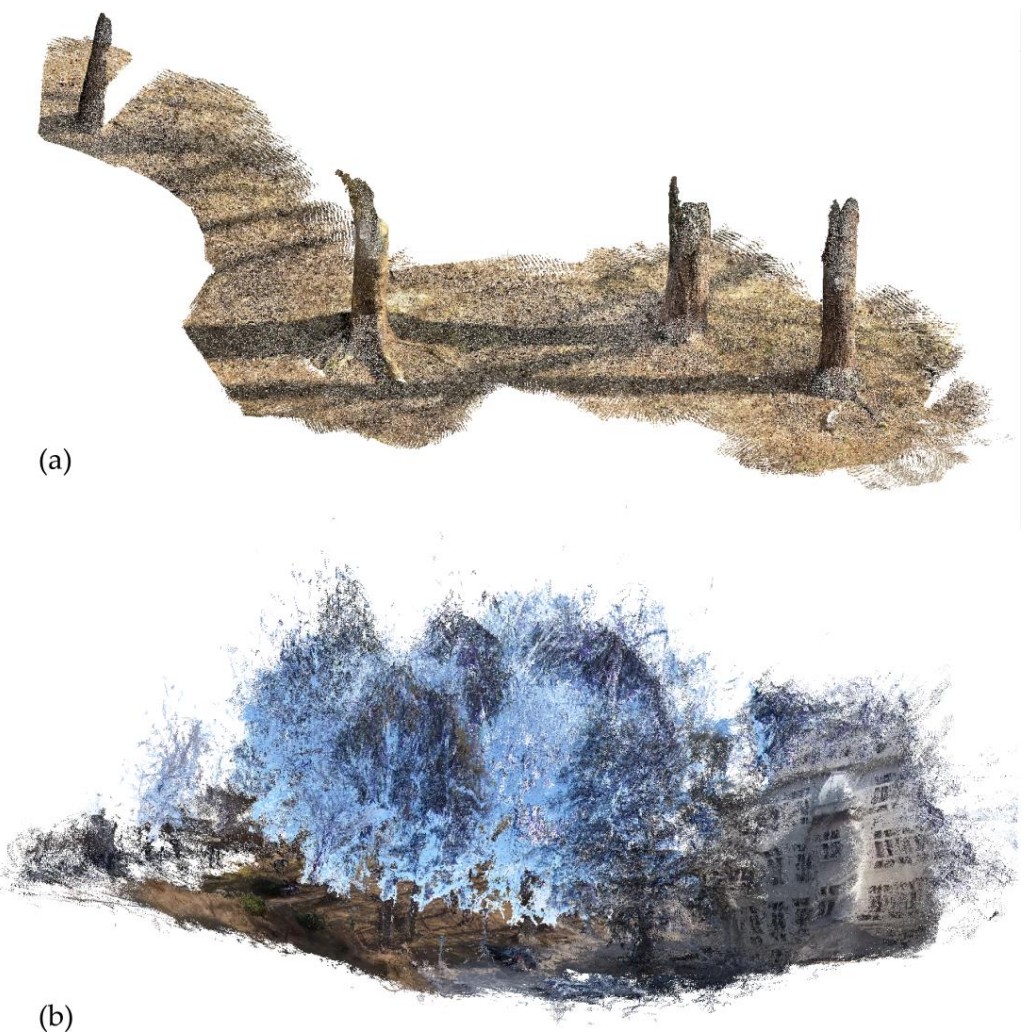

(a)

(b)

**Figure 4.** Results from the winter survey processing: view of the 3D point cloud from one of the iPad scans (**a**); view of the 3D point cloud from spherical photogrammetry (**b**).

### 3.4. Data Elaboration

The last elaboration step with CloudCompare can be divided macroscopically into 'cleaning' (1, 2, and 3) and 'classification' (4 and 5) operations:

1. As a preliminary 'cleaning', a prior attempt to remove the sky's points was made by converting the three RGB channels of the point cloud into a single channel ((R+G+B)/3) (https://www.cloudcompare.org/doc/wiki/index.php/Colors%5CConvert_to_Scalar_Field, accessed on 26 June 2023), so that each point in the cloud was associated with a scalar color between 1 (black) and 255 (white). Filtering by scalar value, we selected a specific range of points: since the sky tends to white, the upper limit of the color range was reduced, resulting in values between 1 and 131 for the winter point cloud, and between 1 and 180 for the summer cloud. In addition to this preliminary selection, we also carried out an RGB color filtering using the Colorimetric Segmenter plugin (https://www.cloudcompare.org/doc/wiki/index.php/Colorimetric_Segmenter_(plugin), accessed on 26 June 2023). By filtering the points for arbitrary RGB color ranges, through several iterations, we were able to select and eliminate points with sky-like RGB values.

2. To further polish the point clouds, the SOR (Statistical Out Removal) filter was used to remove noise (i.e., the elimination of isolated points) based on the calculation of the

average distance of a point from its neighbors (https://www.cloudcompare.org/doc/wiki/index.php/SOR_filter, accessed on 26 June 2023) [39].

3. Finally, to facilitate the acquisition of forest parameters, we manually removed all the context surrounding the study plot (Figure 5).

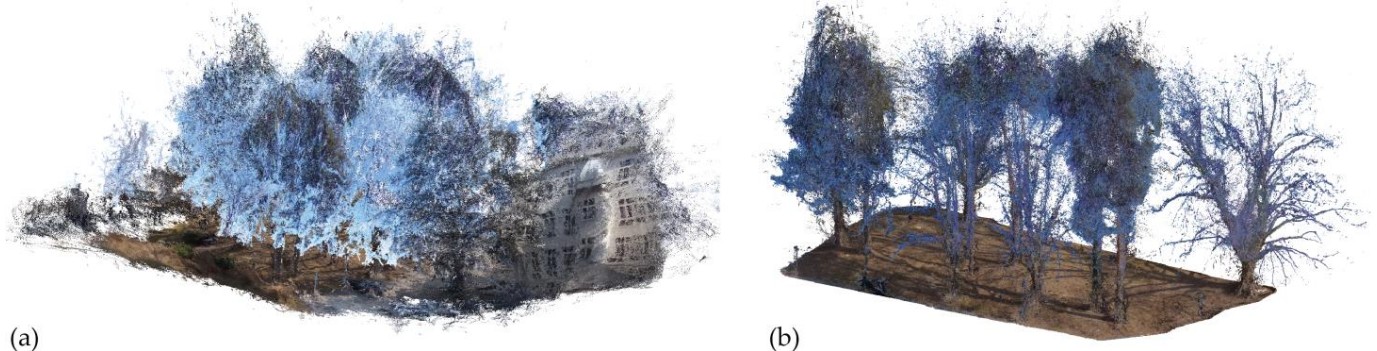

(a)

(b)

**Figure 5.** Views of the 3D point cloud 'winter' model before (**a**) and after (**b**) the 'cleaning' operations.

4. The automatic CSF (Cloth Simulation Filter) filtering algorithm was then applied to separate the above-ground vegetation from the ground points (https://www.cloudcompare.org/doc/wiki/index.php/CSF_(plugin), accessed on 26 June 2023) [39,41] (Figure 6a).

5. Finally, as an example of a procedure that can be repeated for each tree in the plot, one Pinus (tree G) and one Platanus (tree A) were manually isolated from the other trees, lightly cleaned through manual segmentation, and then segmented using the CANUPO plugin (https://www.cloudcompare.org/doc/wiki/index.php/CANUPO_(plugin), accessed on 26 June 2023) in two classes: the first one including trunk and main branches, and the second one with secondary branches and leaves [39] (Figure 6b).

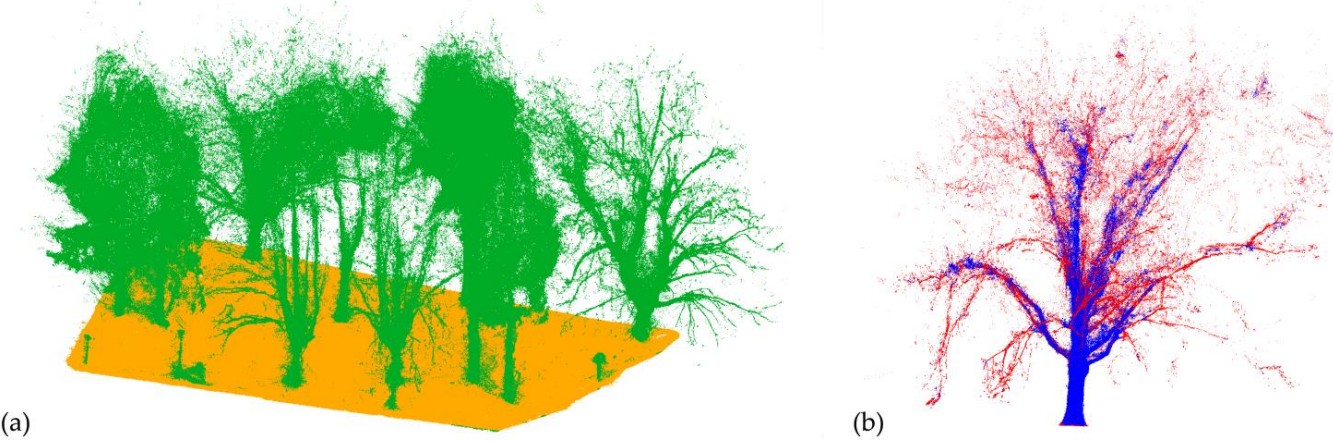

(a)

(b)

**Figure 6.** Left (**a**), results obtained by applying CSF on the entire plot to separate ground points (in orange) and off-ground points (in green); and right (**b**), results of the classification with CANUPO on tree A to distinguish trunk and main branches (in blue) from secondary branches and leaves (in red).

**Metrics data extraction.** Finally, it was possible to derive from the clean and segmented 3D point cloud models of trees A and G, as paradigmatic cases, the following relevant parameters for forestry applications from which various other forest-related values may be derived. At the plot scale:

- Percentage of canopy cover: It was possible to derive information about the canopy, both from winter and summer clouds, using the cross-section tool in CC and extracting

the contours of the canopy and soil as polylines [42,43]. The polylines were then exported as DXF files and imported into Autodesk AutoCAD (https://www.auto desk.it/products/autocad/overview?term=1-YEAR&tab=subscription, accessed on 26 June 2023) for the calculation of the areas covered and uncovered by the canopy (Figure 7a).

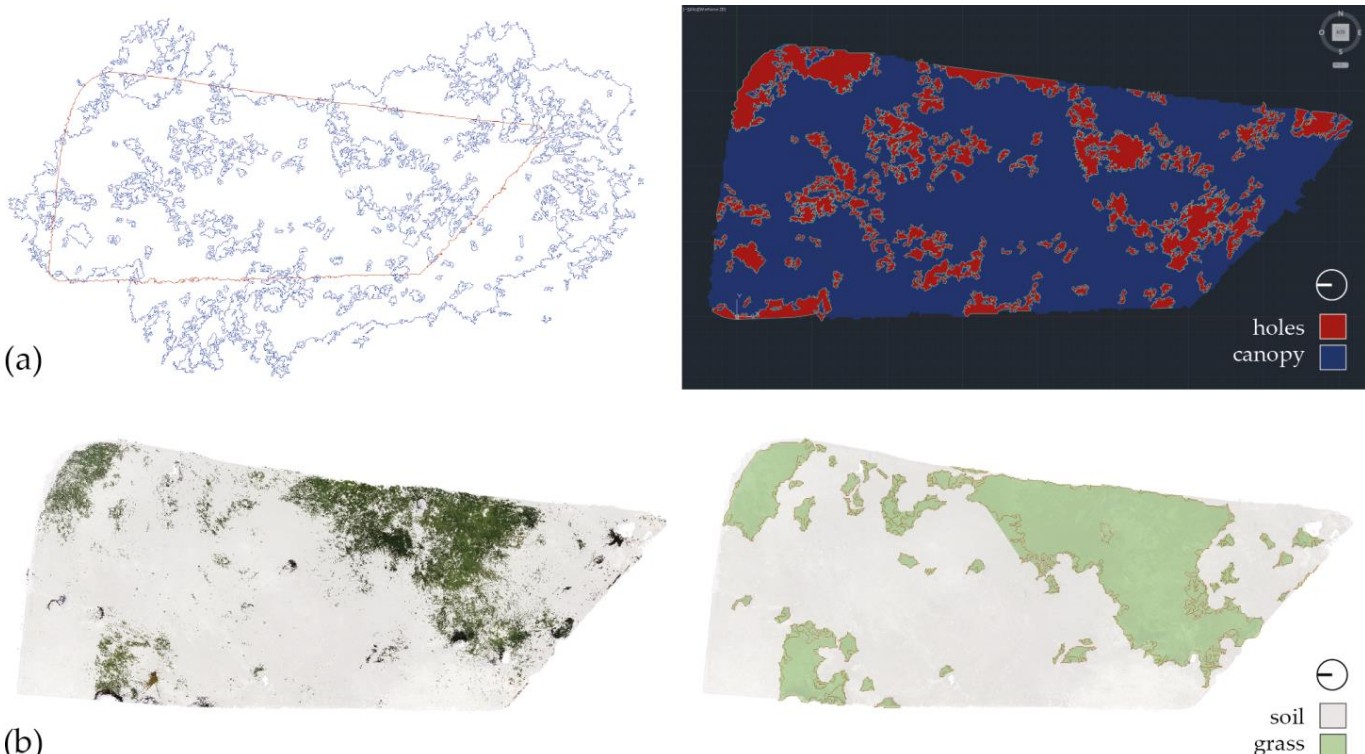

**Figure 7.** Extraction of parameters from the summer point cloud. Top (**a**), screenshots for canopy cover estimation: left, the contours of the canopy and plot extracted from CC, and right, estimation of covered and uncovered areas in AutoCAD. Bottom (**b**), screenshots for grass cover estimation: left, grass points selected with the RGB filter, and right, 2D polygons obtained from the contour of grass areas.

- Percentage of grass cover: For the summer cloud, with the Colorimetric Segmenter plugin and using the RGB filter, we separated grass from soil in CC; then, again, using the cross-section tool, the contours of grass areas were extracted as polylines. Selecting the vertices of all polylines, they can be transformed into 2D polygons for which we computed the total surface (as sum of surfaces) (Figure 7b).

  At the single tree scale (using trees A and G as examples):

- DBH: Using the cross-section tool [32], a slice of the trunk with a thickness of 0.05 m was extracted at 1.30 m above ground; then, the contour was extracted as a polyline and a 2D polygon was created from that [39]. Afterward, two methods were proposed to simulate manual measurements of the DBH (Figure 8a):
- Manual method: Measuring the diameter with the point picking command.
- Mathematical method: approximating the 2D polygon to a circle and knowing the surface of the polygon, it is possible to mathematically derive the diameter [32] (Figure 8a).
- Height: Positioned in a front view, the measurement was acquired manually with the point picking command. Particular attention was paid to extract the measurement between the lowest and highest point of the tree, lying on the same vertical axis [39] (Figure 8b).

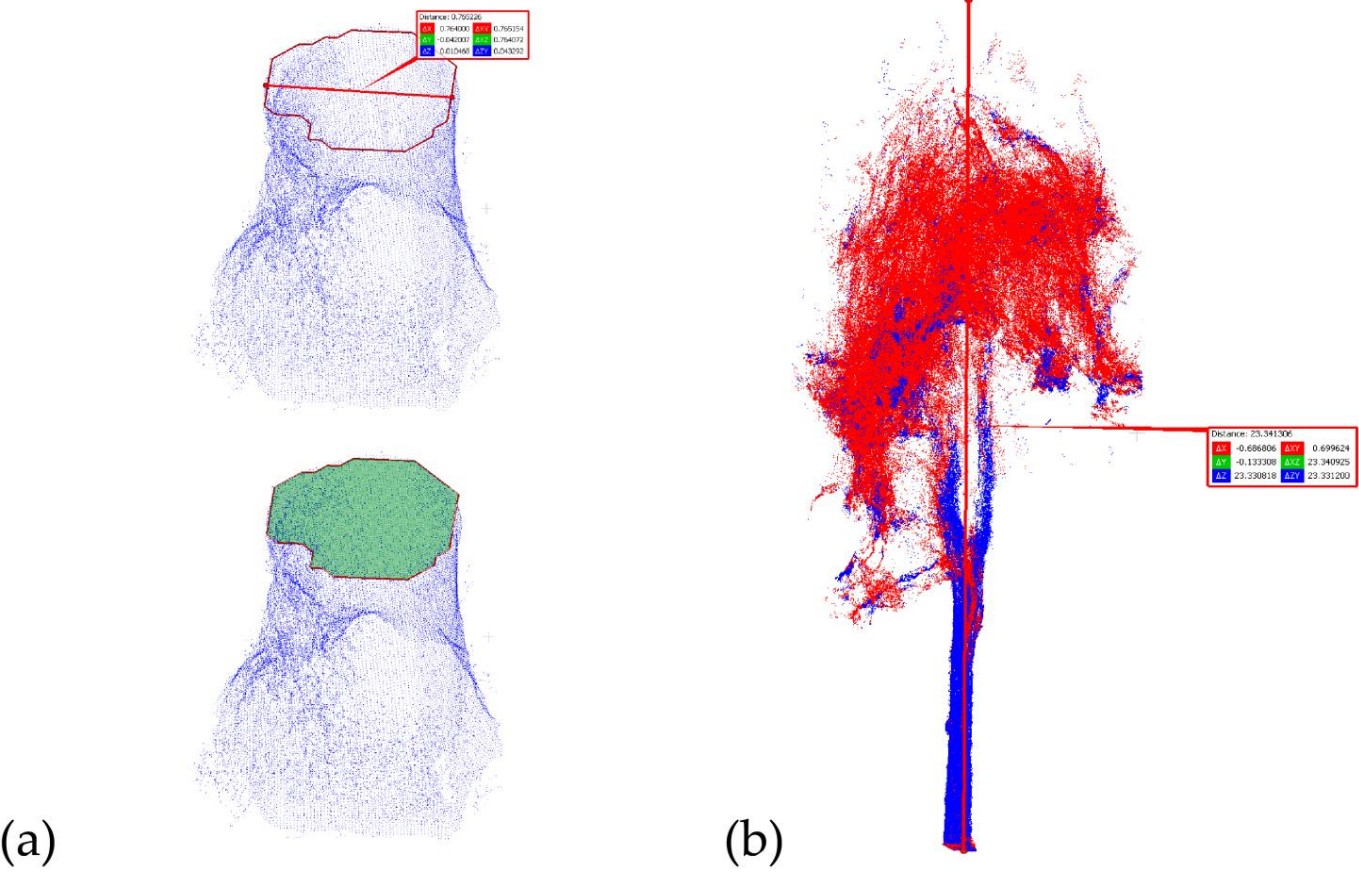

**Figure 8.** Extraction of parameters from the summer point cloud. Left (**a**), screenshots for DBH estimation: top, manual acquisition based on the contours of the trunk section, and bottom, mathematical calculation based on surface of the trunk section. Right (**b**), manual measurements of the tree height.

## 4. Results

At the end of the described workflow—merging survey and digital modeling disciplines and consisting of three main steps (data acquisition, processing, and elaboration)—two (plus one) virtual reconstructions of the selected plot of the Mesiano University park in Trento were obtained: a 3D point cloud of trees up to a height of about 2 m and portions of the surrounding terrain, in the immediate vicinity of the trunks (Figure 9a); and a 3D point cloud of trees up to their canopy and the entire plot's soil, for both winter (Figure 9b) and summer seasons (Figure 9c).

We can derive a qualitative comparison from the tests of the two low-cost sensors used to primarily generate data (Table 2). The acquisition and processing times are significantly shorter (especially with regard to the second operation) using the iPad, but the output information from the latter, as mentioned briefly in Section 3.3, is biased due to the range of its sensor with regard to the detection of elevated elements. On the other hand, the level of detail (meaning geometry and texture) of its scans is on average higher than that of spherical photogrammetry (Figure 10), and the obtained clouds do not need to be scaled (since lidar is an active sensor computing distances with the laser beams), nor georeferenced (thanks to the embedded GNSS, even if its accuracy can be very low). In addition, the used software, plugins, and packages are open-source or have free licensing options (excluding the photogrammetric processing and the metrics data extraction for the plot scale for which proprietary software is actually used), thus also generating open file formats.

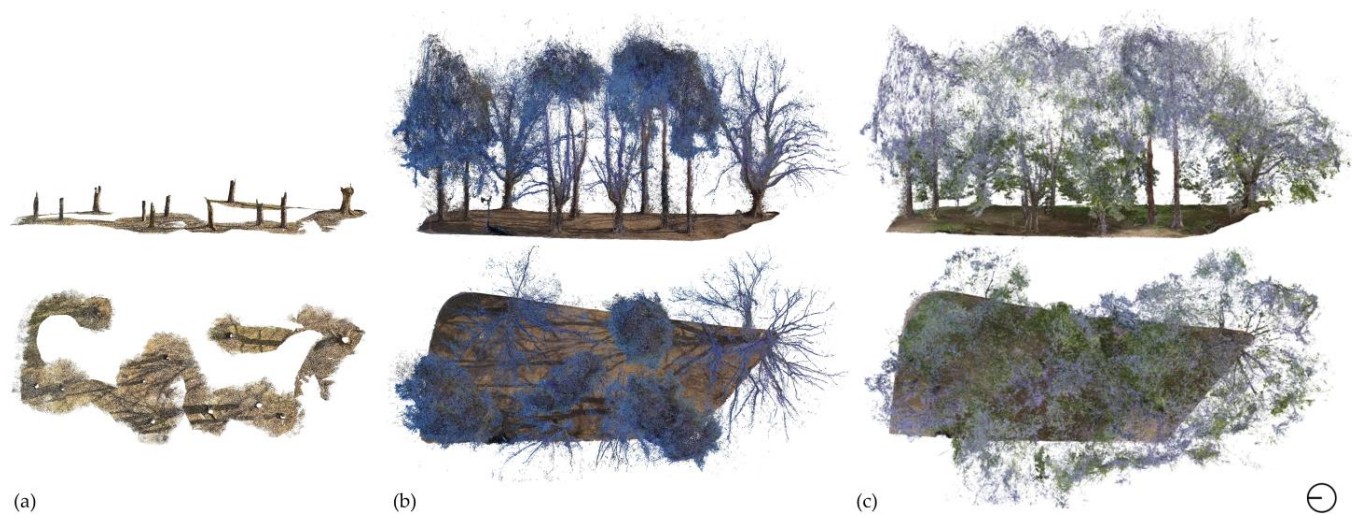

(a)  (b)  (c)

**Figure 9.** Front and top views of the resulting 3D virtual reconstructions: from iPad (**a**), and from spherical photogrammetry with 'winter' data (**b**) and 'summer' data (**c**).

**Table 2.** Qualitative comparison between the two tested low-cost sensors.

| | Apple iPad Pro | Ricoh Theta V |
|---|---|---|
| Cost | €€ | €€€ |
| Acquisition time | ●● | ● |
| Processing time | ●● | ●●●● |
| Used software | - 3D Scanner app<br>- CloudCompare | - RICOH THETA app<br>- Agisoft Metashape Professional [1]<br>- CloudCompare<br>- Autodesk AutoCAD [1] |
| Volume density (mean value) of the resulting point cloud | 13,600 points/r = 0.50 m | 13,900 points/r = 0.50 m |
| 'Automatic' georeferencing of the resulting point cloud | Yes | No |

[1] Commercial software.

With regard to metrics data extraction from the 'complete' 3D models from spherical photogrammetry, the results are summarized in Table 3 for the plot scale and in Table 4 for the tree scale. With respect to the first (Table 3), in our case, the amount of grass area and the percentage of grass cover are derivable only from the summer survey, while for the canopy area and the percentage of canopy cover, a comparison between winter season and summer season is possible and useful. With respect to the second table (Table 4), it can be read by recalling Table 1 and concluding that the data from the traditional survey (e.g., for tree A: DBH 0.77 m, height 21.70 m; for tree G: DBH 0.56 m, height 23.25 m) are consistent with the data retrieved from the digital survey.

**Table 3.** Plot metrics data extraction from 3D models.

| Ground Area (m²) | Season | Grass Area (m²) | Grass Cover | Canopy Area (m²) | Canopy Cover |
|---|---|---|---|---|---|
| 584.43 | Winter | - | - | 385.95 | 0.66 |
| | Summer | 181.40 | 0.31 | 433.93 | 0.74 |

**Table 4.** Single tree metrics data extraction from 3D models.

| Tree | Species | Season | DBH (m) * | Height (m) |
|---|---|---|---|
| A | *Platanus acerifolia* | Winter | 0.75 | 20.28 |
| | Summer | 0.76 | 20.20 |
| G | *Pinus sylvestris* | Winter | 0.60 | 21.39 |
| | Summer | 0.60 | 21.30 |

* The values presented are the average of those obtained using manual and mathematical methods.

The proposed techniques allow for easily repeating the survey and thus for effectively monitoring—even several times a year—the study plot and the trees, leading to a dynamic representation of this urban forest ecosystem by taking into account the change of seasons.

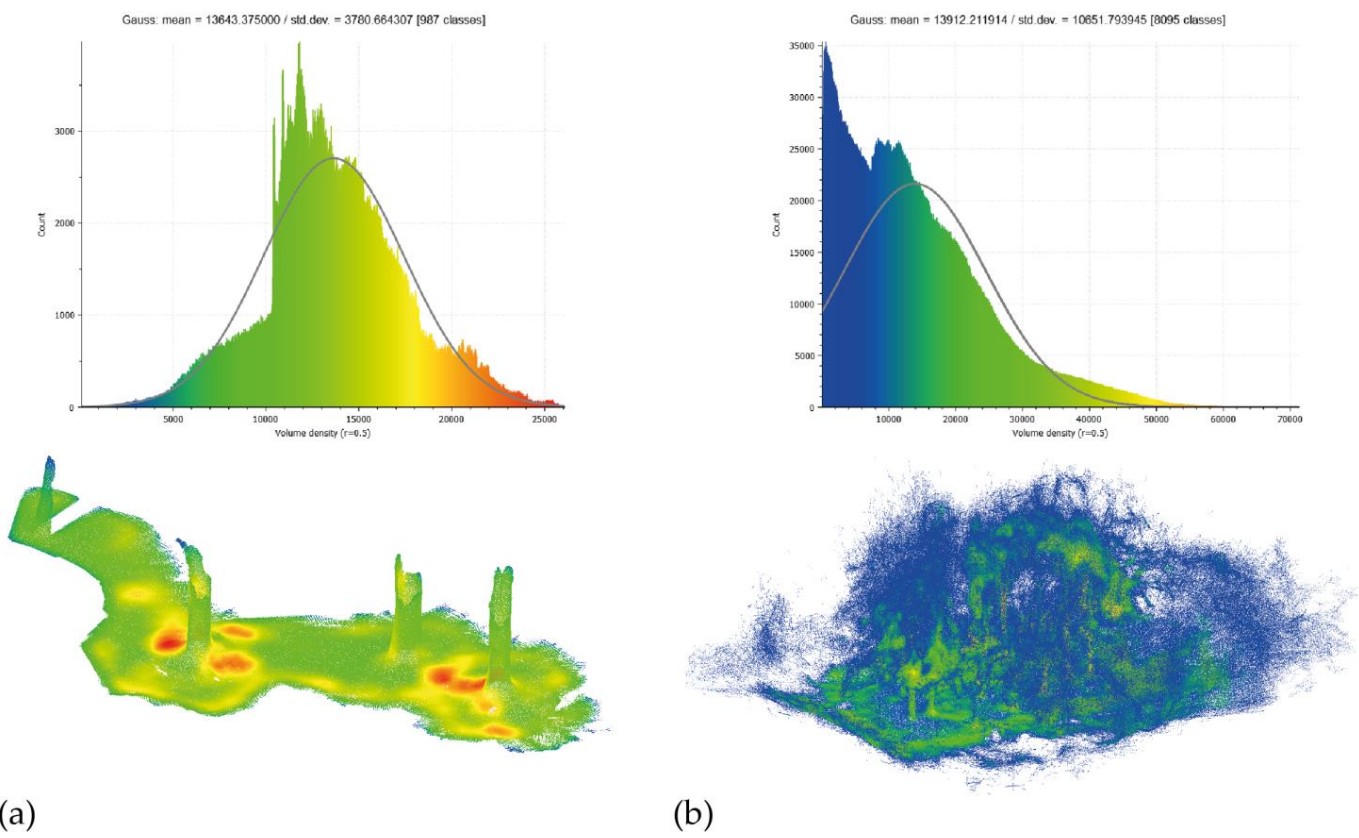

(a)                                        (b)

**Figure 10.** Density of points in a spherical volume of radius 0.50 m: comparison between point cloud from iPad (**a**) and spherical photogrammetry (**b**).

## 5. Discussion and Outlook

The proposed approach illustrates how to acquire multiple 3D data points over time and thus generate digital dynamic models that make possible different visualizations, dependent on the passage of seasons and years, having the possibility of observing and extracting several geometry-dependent parameters that cannot be analyzed in a single timeframe.

The use of Low-cost instruments for data acquisition in forestry proved to be a valid alternative to expensive gold standards, even if the overall geometric accuracy of their outputs is generally lower, and it is also competitive with the traditional models when considering acquisition times, while the same cannot be said when considering data processing and elaboration times, especially regarding the spherical photogrammetric procedure. Another limitation related to the used sensors is linked to their restricted ranges

of acquisition and to their limited memory capabilities; these require the careful planning of the survey in advance in order to 'divide' the study area in portions that have to be acquired in different steps; in addition, the different scans or spherical videos need to be 'connected' later with subsequent possible alignment issues. For these reasons, the selected sample had to be limited in extension even if it is still consistent with the standard dimensions for forest surveys.

With regard to the 'completeness' of the resulting 3D models, there are some shortcomings that cannot always be overlooked: first, by operating from the ground, it is not possible to acquire tree top data accurately enough, especially in the spring and summer period when the canopy is lush; second, there is some difficulty in eliminating sky points when using the spherical photogrammetry method. If the images are not pre-processed, we obtain a point cloud that is chromatically and geometrically greatly altered by the sky points, and when we filter it, we risk losing relevant information.

The implications and potentialities of this study have several future directions, such as the development of an urban virtual forest (i.e., forest digital twins), envisioning the addition of metadata to these digital models, and the interactive sharing of them not only with relevant stakeholders but also with the broader audience. Real-world 3D point clouds of urban forest plots can indeed benefit from the current development of eXtended Reality (XR) applications, using game engines such as the free Unity; to make the access of technical information more intuitive and interactive, recent studies experimented with (large) point clouds rendering in VR [44,45] not only for research and academic purposes as carbon sequestration estimation or monitoring biodiversity [46], but also for engaging and disseminating applications, such as allowing the 'sensing' of other dimensions such as sound [47,48].

Filling the lack of understanding about methodologies and tools on the one hand, and of standardized procedures on the other hand, designers, architects, and landscape architects have the potential to handle the essential stages of managing information regarding urban trees and green spaces—which include acquiring and processing data, analyzing it, and visualizing the results—to support their design work. It would be interesting to understand how to optimize the work process and further reduce processing times for large amounts of data (as in the case of monitoring road trees in large cities).

Finally, procedures such as the one presented here could also acquire a participatory characteristic by considering the general audience as prosumers (both producers and consumers) of data and information about urban green infrastructure; professionally generated 3D digital reconstructions could theoretically be fueled and periodically updated with videos, photos, lidar scans from mobile devices, etc., collected and shared by citizens and tourists, who are becoming increasingly involved in and responsible for the management of public green spaces.

**Author Contributions:** Conceptualization, C.C., A.M., A.P., M.C., S.F. and G.A.M.; methodology, C.C., A.M. and A.P.; software, C.C. and A.M.; formal analysis, C.C. and A.M.; investigation, C.C., A.M. and A.P.; data curation, C.C. and A.M.; writing—original draft preparation, C.C., A.M. and A.P.; Section 1, C.C.; Section 2.1, A.P.; Section 2.2, C.C.; Section 3.1, A.P.; Section 3.2, A.M.; Section 3.3, C.C.; Section 3.4, A.M.; Section 4, C.C. and A.M.; Section 5, C.C., A.M. and A.P.; writing—review and editing, M.C., S.F. and G.A.M.; visualization, C.C. and A.M.; supervision, M.C., S.F. and G.A.M. All authors have read and agreed to the published version of the manuscript.

**Funding:** This research received no external funding.

**Institutional Review Board Statement:** Not applicable.

**Informed Consent Statement:** Not applicable.

**Data Availability Statement:** The data presented in this study are available on request from the corresponding author.

**Acknowledgments:** The authors acknowledge the Laboratory of Architectural Modelling and Analysis Representation and Communication (LAMARC) from the University of Trento-DICAM for

technical and instrumental support at all stages of this research, Alfonso Vitti from the University of Trento-DICAM for sharing his knowledge about terrestrial laser scanning, and Arnadi Murtiyoso from ETH-FORM and Hristina Hristova of the WSL Switzerland for their help in understanding the 360° camera's data processing. The authors would also like to thank Chiara Frungillo and Andrea Biotti from the University of Trento-DICAM for their operative contributions in the data acquisition in February 2023.

**Conflicts of Interest:** The authors declare no conflict of interest.

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
