# Peer review of "Low-Cost 3D Virtual and Dynamic Reconstruction Approach for Urban Forests: The Mesiano University Park"

_sustainability, doi:10.3390/su151914072_

Round 1

Reviewer 1 Report

The authors of this contribution propose a low-cost and fast workflow for 3D virtual reconstructions of the trees in the urban context. The question is of great interest and usefulness for planners, forestry operators and for public administrations that have to manage the arboreal heritage present in the cities, having to acquire data relating to trees in an evolutionary dynamic.

The structure of the article is very clear. The introduction presents an overview of the traditional survey framework and briefly introduces the current issue of 3D data in forestry. Section 2, after presenting the case study, describes methods, tools, and software used in the data acquisition, processing, and elaboration steps of the proposed workflow.

Several methods have been used for phase surveying, including spherical photogrammetry and point cloud modelling; and several tools were tested, including the use of a 360° camera, tablet with lidar sensor. The data were processed and elaborated through a "cleaning" operation (has been eliminated points with sky-like RGB values and has been manually removed all the context surrounding the study plot); and classification to distinguish trunk and main branches from secondary branches and leaves of the trees.

The result obtained from this experimental process is a three-dimensional model, which makes the geometry and morphology of the trees very clearly distinguishable.

In the discussion and conclusions both the strengths and limitations of the proposed workflow were highlighted, which in any case requires a laborious procedure for data processing.

The paper could also be accepted in its current form. Any case, I would like to suggest the authors to further develop the discussion relating to the future developments of this experimentation.

The use of fast techniques for acquiring and processing geometric and morphological data of trees could be particularly useful for monitoring road trees. Suffice it to consider that a city like Rome or Paris, has around 120,000  street line trees. It would be interesting to understand how to optimize the work process and further reduce processing times for large amounts of data.

Author Response

Thank you for your positive comments.
We addressed in the manuscript your suggestion to further develop the discussion section, including outlooks of this research experimentation.

Reviewer 2 Report

Thanks a lot for authors' contributions to urban forests research. The authors apply low-cost 3D virtual and dynamic reconstructions approach in the study. They take the Mesiano University Park as the case study. This is an interesting research but there are still some comments as follows:

1. In abstract, please clarify the research gaps.

2. The manuscript's structure is lack of literature review section, please transfer some contents in introduction, such as setcion 1.1,  1.2, etc. to literature review part.

3. Please clarify the limitations and further research directions in the conclusion part.

Reviewer 3 Report

The article "Low-cost 3D virtual and dynamic reconstructions approach for urban forests: the Mesiano University Park" presented for review is interesting and refers to the currently intensively developing research direction, modern technology in forestry. The authors specified the research methodology quite precisely. Nevertheless, other chapters require some additions.

The title mentions "low-cost" but the content does not explain what it is and why we consider these technologies to be low-cost.

The introduction describes data obtained in a traditional way from the forest environment, but these data (habitat type, health status, ecological functions, etc.) cannot be obtained from the method presented later. It can better describe for what purposes the data can be obtained.

The first paragraph in the chapter "2. Materials and Methods" is probably the goal, although the introduction also states the goal of the article, or rather the expose (which should be changed). The Discussion chapter talks about a hypothesis that has not really been put forward before.

Results: it seems that table 3 should include at least two sentences of the description of the results.

Limitations on the use of the method should be included in the discussion. For example, what can be the maximum density, crown cover, area, age.

Reviewer 4 Report

Regarding the authors' conclusion, it seems that the results of the current study indeed support the initial hypothesis, which suggested the feasibility of associating forest inventory with cartographic activity. However, I would like to emphasize certain concerns that arise from a more detailed analysis of the work.First and foremost, it is noteworthy that the study was based on an extremely small sample, consisting of only eleven trees, evaluated at two different moments. Such a limited sample size raises questions about the representativeness of the obtained results, especially when considering the vast diversity present in forest environments. The utilization of a method applied to such a confined area raises doubts about extrapolating these results to more extensive forest ecosystems.

Another point of concern is the absence of solid scientific underpinning for the employed method. The use of modern technologies, such as scanning digitization, is undoubtedly promising, but it is essential for such an approach to be grounded in established theories and previously validated methods. Without a robust scientific foundation, the presented method may lack reliability and reproducibility, undermining the credibility of the results and conclusions. Furthermore, it is valid to question the applicability of the findings of this study in broader contexts. The growth rate of the investigated species may be subject to a range of local factors, such as climatic conditions, resource availability, and ecosystem interactions. Without an in-depth analysis of these aspects, it is challenging to justify generalizing the results as an accurate estimate of forest growth.Considering these points, doubt arises about the relevance of the article to the scientific community and forest management practice. Although the method may hold some utility as a preliminary growth estimate for the studied species, the discussed elements cast doubt on the feasibility of its widespread application.

Based on these considerations, I am inclined to agree with the opinion that this article may not be suitable for publication.

Round 2

Reviewer 3 Report

the authors commented on the issues raised

Reviewer 4 Report

Despite my initial opinion about the article being inclined towards rejecting its publication, I have to admit that the authors made substantial changes and satisfactorily justified the criticisms raised by this reviewer. After a thorough reading of the latest version, I am inclined to support the publication of this manuscript.